# Counting Mixed Breeding Aggregations of Animal Species Using Drones: Lessons from Waterbirds on Semi-Automation

**Roxane J. Francis** *, **Mitchell B. Lyons**, **Richard T. Kingsford** and **Kate J. Brandis**

Centre for Ecosystem Science, University of New South Wales, Sydney NSW 2052, Australia;
Mitchell.Lyons@unsw.edu.au (M.B.L.); Richard.Kingsford@unsw.edu.au (R.T.K.);
Kate.Brandis@unsw.edu.au (K.J.B.)
* Correspondence: roxane.francis@unsw.edu.au

**Abstract:** Using drones to count wildlife saves time and resources and allows access to difficult or dangerous areas. We collected drone imagery of breeding waterbirds at colonies in the Okavango Delta (Botswana) and Lowbidgee floodplain (Australia). We developed a semi-automated counting method, using machine learning, and compared effectiveness of freeware and payware in identifying and counting waterbird species (targets) in the Okavango Delta. We tested transferability to the Australian breeding colony. Our detection accuracy (targets), between the training and test data, was 91% for the Okavango Delta colony and 98% for the Lowbidgee floodplain colony. These estimates were within 1–5%, whether using freeware or payware for the different colonies. Our semi-automated method was 26% quicker, including development, and 500% quicker without development, than manual counting. Drone data of waterbird colonies can be collected quickly, allowing later counting with minimal disturbance. Our semi-automated methods efficiently provided accurate estimates of nesting species of waterbirds, even with complex backgrounds. This could be used to track breeding waterbird populations around the world, indicators of river and wetland health, with general applicability for monitoring other taxa.

**Keywords:** UAV; machine learning; colony; open source; GIS; avian; remote sensing; heronry

## 1. Introduction

There is an increasing need to estimate aggregations of animals around the world, including turtles, seals and birds [1–6]. Regular monitoring of these concentrations allows decision-makers to not only track changes to these colonies but also long-term environmental changes, given that large aggregations of some species can be used to monitor environmental change (e.g., waterbird breeding colonies) [7,8]. Existing methods to monitor such occurrences include the use of camera traps [9,10], radar [11], aerial surveys [12,13] and in-situ observers [14,15]. Each of these methods has limitations, including expense [9], poor accuracy [16] or disturbance to wildlife [14].

Drones, or unmanned aerial vehicles (UAVs), can collect considerable data quickly over large areas. They provide advantages over in-situ observations, accessing physically inaccessible or dangerous areas in a relatively small amount of time [17–19]. Drones are also relatively cheap, safe and less disturbing, improving traditional wildlife surveys [3,18,20,21]. They can, however, disturb some animal populations, requiring careful consideration of appropriateness when surveying [22]. As a result of such time and cost savings, drones are increasingly used to monitor bird communities [23–25].

Alongside the increasing availability of large amounts of drone datasets, there is a need for effective and efficient processing methods. There are two broad options: manual counting of images and

semi-automated methods. The former can be extremely labour-intensive and consequently expensive, particularly for large aggregations of wildlife [26], further complicated when more than one species is counted. Semi-automated methods, including the counting of animals from photographs (e.g., camera traps) and drone imagery, are increasingly being developed around the world [27]. These methods reduce the time required to count and process drone images [28], accelerating the data entry stage and encouraging the use of drones as scientific tools for management. Such benefits allow for real-time monitoring and management decisions and could, for example, assist in the targeted delivery of environmental flows for waterbird breeding events [29].

Generally, semi-automated counting methods are most effective for species where there are strong contrasts against the backgrounds, particularly when background colours and shapes are consistent [28]. They can distinguish large single species aggregations on relatively simple backgrounds [30–32], up to sixteen avian species (numbering in the hundreds) on simple single colour backgrounds, such as oceans [33,34], or single species aggregations of hundreds of thousands on complex backgrounds [3].

Development of flexible, repeatable and efficient methods, using open source software, is important in ensuring methods are applicable across a range of datasets [35,36]. Further, there are potential cost implications of processing data, given that some processing software can be expensive (i.e., compulsory licence fees, called 'payware' in this paper) and so are often only accessible to large organisations in high-income countries [37]. Open source software, or software with optional licence/donation fees ('freeware' in this paper), can overcome such restrictions, providing repeatable processing techniques, which are accessible to all users.

We aimed to develop a semi-automated method for counting large aggregations of mixed species of breeding waterbirds, with highly complex vegetation backgrounds. Specifically, we had four objectives: (1) to develop a transferrable semi-automated counting method with high accuracy (>90%) for counting mixed species of breeding colonies on complex backgrounds, (2) to compare the time using a semi-automated compared to a manual method, (3) to identify whether birds were on (incubating) or off their nests and (4) to ensure methods were reproducible and accessible by comparing two processing pathways (freeware to payware). Finally, we discussed such an application on other breeding aggregations of wildlife.

## 2. Materials and Methods

### 2.1. Study Areas

We focused on two different waterbird breeding colonies (Figure 1): the Kanana colony in the Okavango Delta, Botswana, and the Eulimbah colony in the Lowbidgee floodplain, Australia. The colonies were respectively established in 2018 and 2016 following flooding, in a range of vegetation types (Table 1).

**Table 1.** Main waterbird breeding species (targets) in the two waterbird colonies, Kanana colony (Okavango Delta) and Eulimbah colony (Lowbidgee floodplain), counted using semi-automated methods, including their size and colour (important for detection), the dominant vegetation on which they nested (the background) and estimated number of each species in the two colonies.

| Colony | Waterbird Descriptions | | | Dominant Vegetation |
| | Species | Colour | Size (cm) | |
|---|---|---|---|---|
| Kanana | African Openbill *Anastomus lamelligerus* | Black | 82 | Gomoti fig *Ficus verrucolosa* |
| | African Sacred Ibis *Threskiornis aethiopicus* | White | 77 | Papyrus *Cyperus papyrus* |
| | Egret sp. *Egretta sp* [1] | White | 64–95 | |
| | Marabou Stork *Leptoptilos crumeniferus* | Grey | 152 | |
| | Pink-backed Pelican *Pelecanus rufescens* | Grey | 128 | |
| | Yellow-billed Stork *Mycteria ibis* | White | 97 | |
| Eulimbah | Australian White Ibis *Threskiornis molucca* | White | 75 | Lignum shrubs *Duma florulenta* |
| | Straw-necked Ibis *Threskiornis spinicollis* | Grey | 70 | Common reed *Phragmites australis* |

[1] Predominantly Yellow-billed Egrets *Egretta intermedia* with some Great Egrets *Ardea alba*.

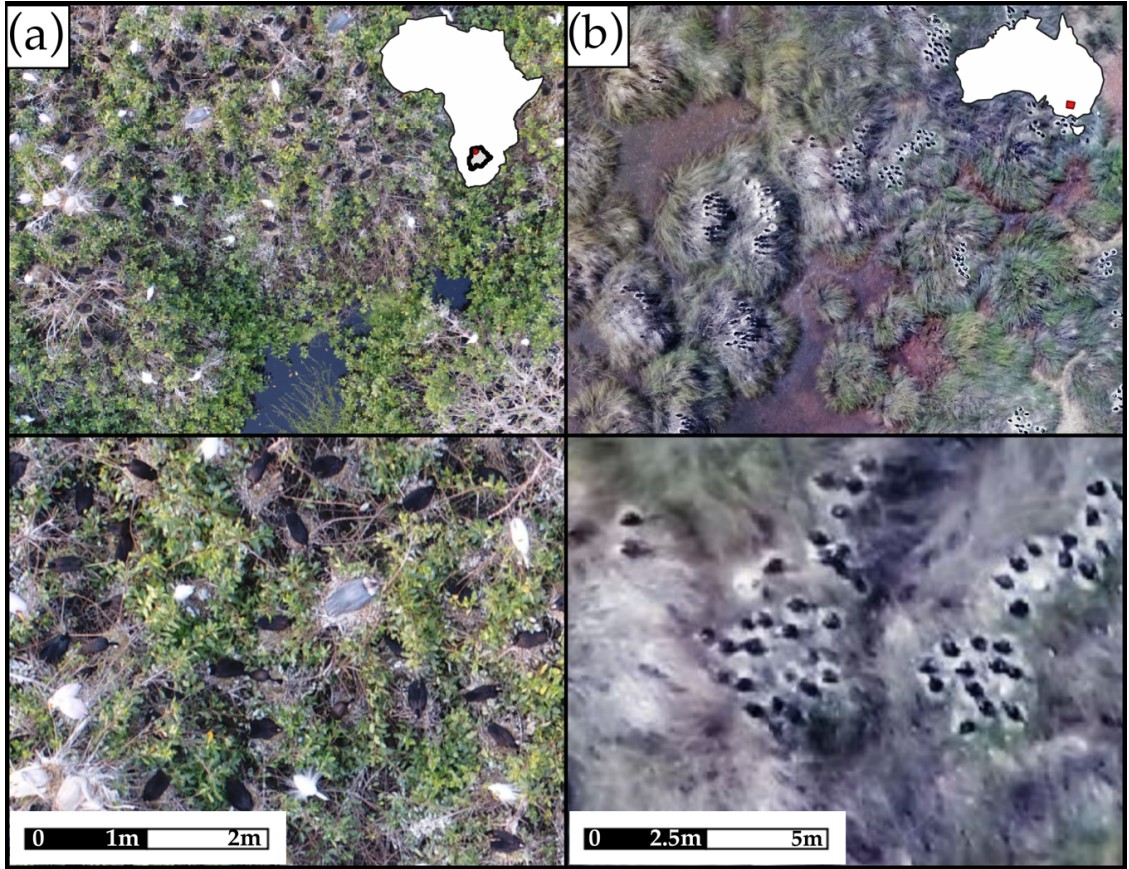

**Figure 1.** Locations' imagery at two resolutions and an example of the segmentation process of the two waterbird colonies: (**a**) Kanana (Okavango Delta, Botswana) taken using a Phantom 4 Advanced at 20 m, and (**b**) Eulimbah (Lowbidgee floodplain, Australia) using a Phantom 3 Professional at 100 m.

## 2.2. Image Collection and Processing

First, we created polygons surrounding the Kanana colony in September 2018, using Pix4d Capture [38], allowing pre-programming of drone flights and reducing drone noise by adjusting the flight's height and speed [24]. We collected imagery using a DJI Phantom 4 Advanced multi-rotor drone with the stock standard 20 MP camera (5472 × 3648 image size, lens Field of View (FOV) 84° 24 mm) over the breeding colony (30–40 ha), at a height of 20 m. We flew the drone at the slowest speed (~2 ms$^{-1}$), with 20% front and side image overlap, taking still images at evenly spaced intervals, along parallel line transects, with the camera positioned at 90° (nadir perspective). Waterbirds mostly remained on their nests. Resulting photos were clipped to remove the 20% overlap on each side and placed into a 5 × 9 grid (Figure 2, Step 1), with images aligned within the freeware Photoscape X [39]. We did not orthorectify the images, treating them as joined images (jpegs), in an arbitrary coordinate system, allowing us to provide a freeware processing pathway.

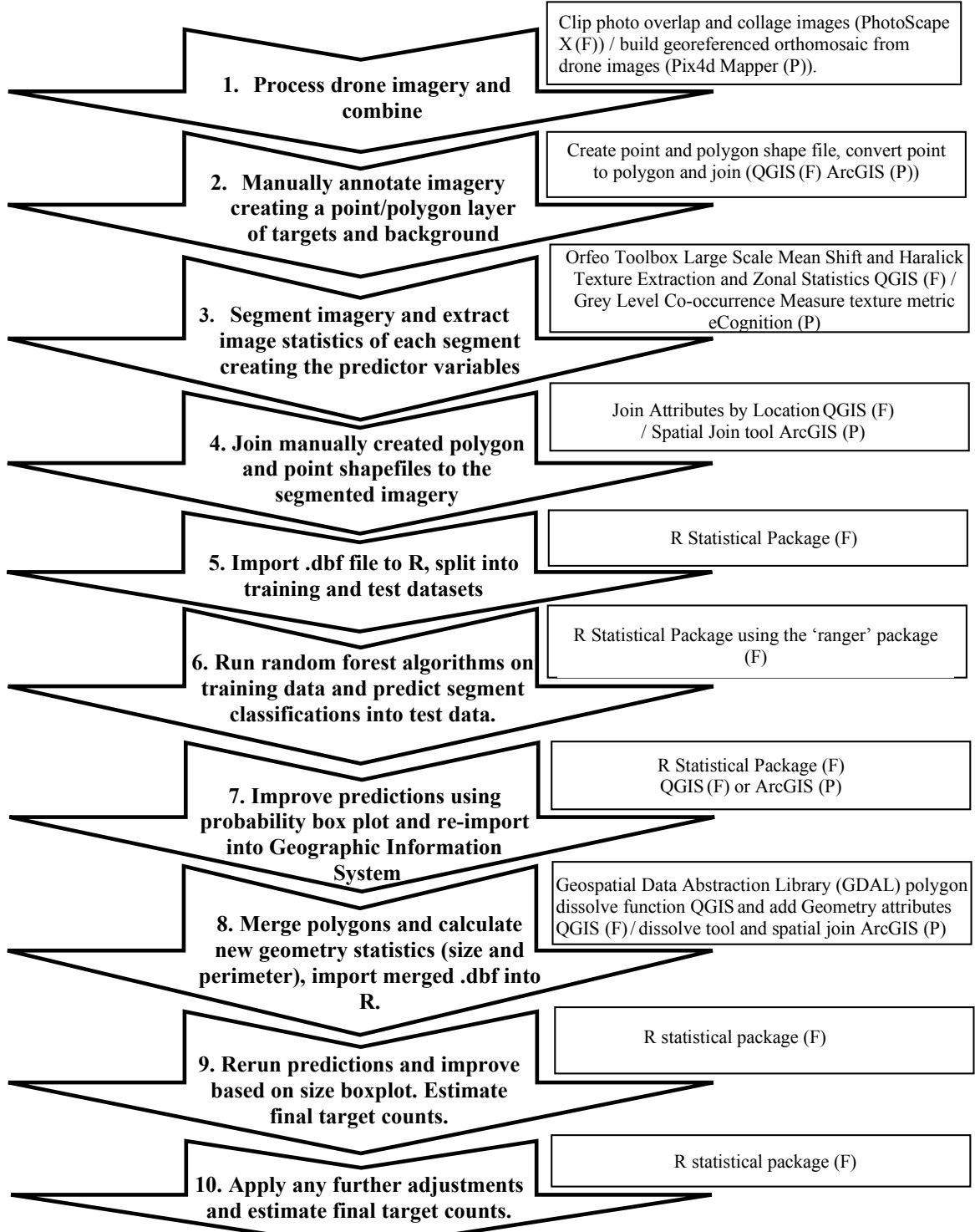

**Figure 2.** The ten steps required to process drone imagery of waterbird colonies using our semi-automated approach, with descriptions of specific software, tool boxes and functions compared (large-scale mean shift (LSMS), freeware (F) and payware (P)).

We flew the Eulimbah colony manually in October 2016 at a height of 70 m, launching the drone from a nearby levee bank to reduce disturbance, given that many birds were moving on and off their nests. We collected imagery over the colony (15–20 ha) using a DJI Phantom 3 Professional multi-rotor drone, again with the stock standard camera and an additional neutral density filter (4000 × 3000 image

size, lens FOV 94° 20 mm). We flew at 5–10 ms$^{-1}$ aiming to acquire imagery with ~70% forward and lateral overlap, along parallel flight lines at 90° [3,25]. We processed the imagery using the commercial software Pix4DMapper (v4.19,166 Pix4D SA), with a photogrammetry technique called 'structure from motion', which identified points in overlapping images, building a three-dimensional (3D) point cloud reconstruction of the landscape, and finally, generating a digital surface model and an orthorectified image mosaic (Figure 2, Step 1). This data was originally collected for another purpose, hence the differing collection methods between colonies, however this allowed us to test the transferability of the following methods.

*2.3. Semi-Automated Image Analysis*

We aimed to develop transferable methods for the two datasets, despite different data collection methods (drone, height), colonies (locations, species) and image processing pathways. We delineated targets (waterbird-related) and backgrounds (surrounding areas in the colony). There were five target species in the Kanana colony (Yellow-billed Storks *Mycteria ibis*, African Openbills *Anastomus lamelligerus*, Marabou Storks *Leptoptilos crumeniferus*, egrets (predominantly Yellow-billed Egrets *Egretta intermedia* and some Great Egrets *Ardea alba* which could not be separated) and Pink-backed Pelicans *Pelecanus onocrotalus*) and two species in the Eulimbah colony (Straw-necked Ibis *Threskiornis spinicollis*, Australian White Ibis *Threskiornis Molucca*). At the Eulimbah colony, we also separately identified whether the two species were on-nests or off-nests (Straw-necked Ibis only), or if the nest had egg/s or was just nest material: in total, five target objects at each colony.

We used a supervised learning approach, given the complexities of the mixed species' aggregations and varied background vegetation. This included an object-based image analysis [40] and a random forest machine learning classifier [3]. The approach had five steps: (1) curation of a training and test dataset (subsets of the entire dataset) for respective modelling and validation, (2) segmentation of the image data (entire dataset) into objects for modelling, with the extraction of colour, texture and morphological features of image objects to use as predictors, (3) fitting of a random forest model to predict different target objects into images across the entire datasets and (4) estimation of target species' numbers in the two colonies.

### 2.3.1. Training and Test Datasets

Supervised machine learning required a training dataset to develop the algorithm and a test dataset for targets (one for each colony), before estimating target species numbers in the colonies. We therefore manually annotated up to 50 of each target object including birds and nests (where possible) on the original imagery, incorporating a range of different images and areas of the colony (Figure 2). We also delineated enough 'background' polygons (5-10 in each colony) to include the range of different backgrounds visible (e.g., water, vegetation, bare ground, sand and mud) to train the algorithm, allowing for their specification as non-targets, producing point (targets) and polygon (background) shapefiles (Figure 2, Step 2).

### 2.3.2. Image Object Segmentation and Predictor Variables

For these two (one for each colony) manually selected datasets of targets and backgrounds, we combined object-based segmentation principles, grouping similar attributes (texture, shape, neighbourhood characteristics [41]), with machine learning predictive modelling for semi-automated detection of birds from drone imagery [40,42]. We compared two image segmentation approaches on each image set from the Kanana and Eulimbah colonies: orfeo toolbox in QGIS v3.6.3 (freeware) and eCognition v8 (payware) (Figure S1). We used trial and error for the spatial radius parameters, starting with the defaults and adjusting based on visual determination of appropriate segment size, ensuring effective delineation of individual birds/targets. This resulted in 20 for the Kanana colony and 100 for the Eulimbah colony, reflecting differences in pixel size (smaller pixels and lower height in the Kanana colony) (Figure 2, Step 3). Each image segment was attributed with its colour properties

(brightness, mean and standard deviation of the blue, green and red photo bands, Figure 1), geometric features (e.g., size, ellipse radius, rectangularity, roundness), and textural character (e.g., Gray-Level Co-Occurrence Matrix (GLCM) contrast, entropy), depending on the software used (Figure 2, Step 3).

After segmentation, the manually created point and polygon files of targets and background were then intersected with the corresponding segmented image layer (Figure 2, Step 4), separately using the freeware and payware. As a result, each target object and/or background segment was associated with its corresponding suite of predictor variables and exported as database files (.dbf) for import into R for modelling [43].

### 2.3.3. Machine Learning

We developed our machine learning methods in R on the imagery from the Kanana colony. After importing the two .dbf files into R (freeware and payware files), we split the manually identified datasets into training (80%) and testing (20%) groups, using stratified random sampling focused on background and targets (Figure 2, Step 5). We first developed and tested our modelling and classification on these two datasets and then fitted the model to the entire image sets to estimate the total numbers of targets.

On the training dataset, we used the random forest algorithm, a widely used machine learning approach which deals with correlated or redundant predictor data by creating decision trees, where each different split is based on choosing from a random subset of the predictor variables [44]. We fitted separate random forest models to the training dataset of each approach (freeware versus payware), using the 'ranger' package on R v3.4.x [45] (Figure 2, Step 6). First, our classification tree separated different target and background features. We then fitted a (binomial) regression tree, splitting bird and non-bird targets into 1 and 0 respectively, based on the probability of identification as a bird. The random forest classification and regression used 1000 trees, the square root of the number of predictors as the size of the random subset to choose at each split, and a minimum leaf population of 1 for classification and 5 for regression [44,45]. The final prediction models used the mode of the classification trees and the mean of the predictions for our regression trees.

We then tested our prediction models on the test data (remaining 20%), reporting accuracy (Figure 2, Step 6). To improve classification predictions and better separate between the target and background classes, we inspected the data using boxplots and 99% quantile statistics and developed thresholds (Figure 2, Step 7). We changed segments that were likely to have been misclassified, as either bird or background, to the correct class based on the values of the 99% quantile (Figure 3). We reported on comparison of these datasets as a standard error matrix-based accuracy assessment.

The classified database files (.dbf), with target and background probabilities corrected, were reimported into GIS software (using freeware or payware). They were inspected visually, and we noted there were cases where a single target was divided into two segments. We corrected this by merging neighbouring segments, with the same classifications, ensuring that targets were only counted once. We then calculated the new segment area and perimeter and imported the database files (.dbfs) back into R (Figure 2, Step 8). We reran the prediction models and created boxplots of the areas identified for each species (Figure 4), which allowed us to detect outliers in area across both datasets (freeware or payware), that exceeded thresholds as specified by the 99% quantile and which therefore needed to be reclassified as targets or background (Figure 2, Step 9). We replicated the code and GIS steps above and tested transferability of our approach to the Eulimbah colony.

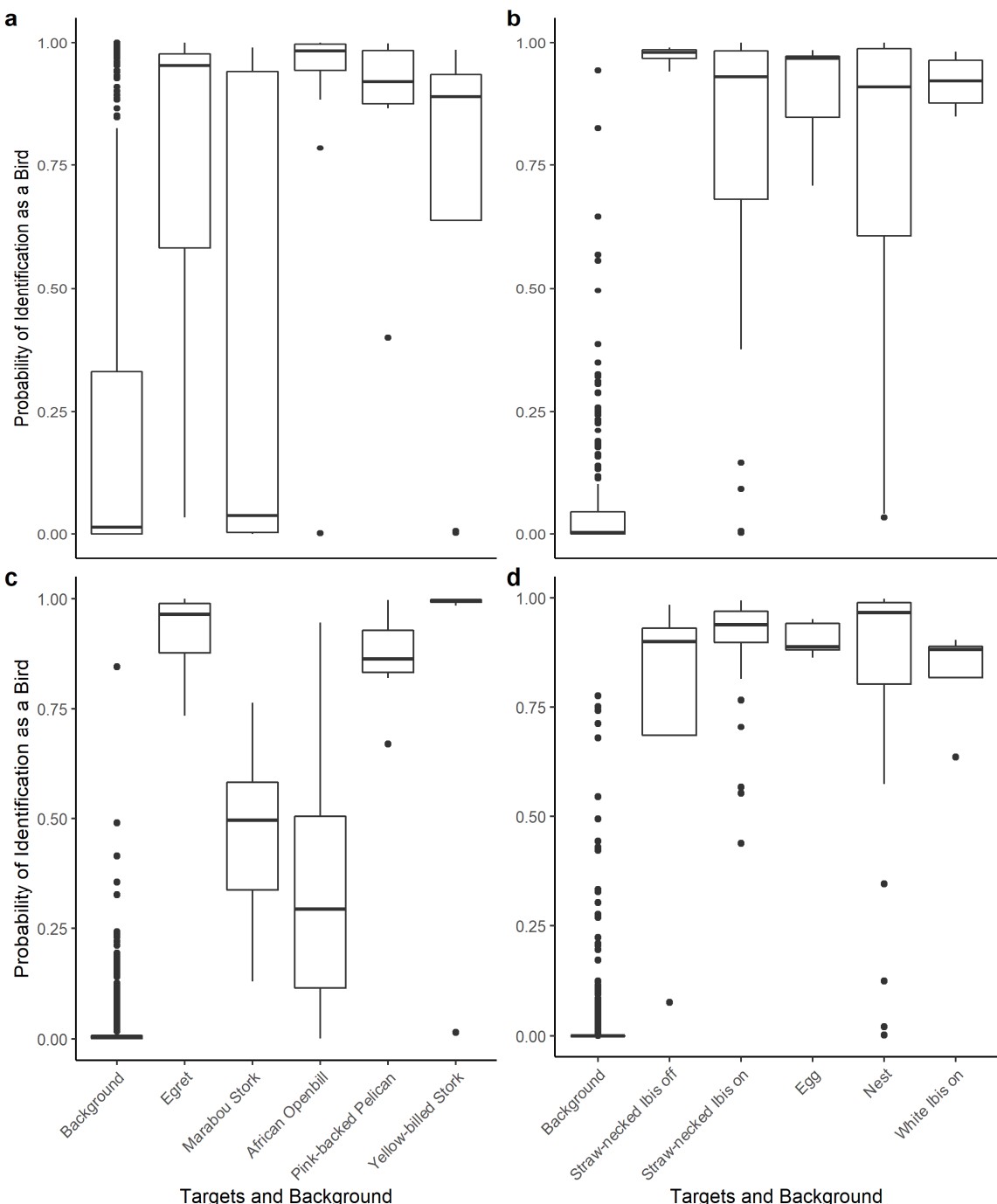

**Figure 3.** The boxplot used to identify classification errors between targets and background using 99% thresholding for the freeware method at (**a**) the Kanana colony and (**b**) the Eulimbah colony, and the payware method at (**c**) Kanana and (**d**) Eulimbah. At the Eulimbah colony, birds were identified as being either on or off their nests.

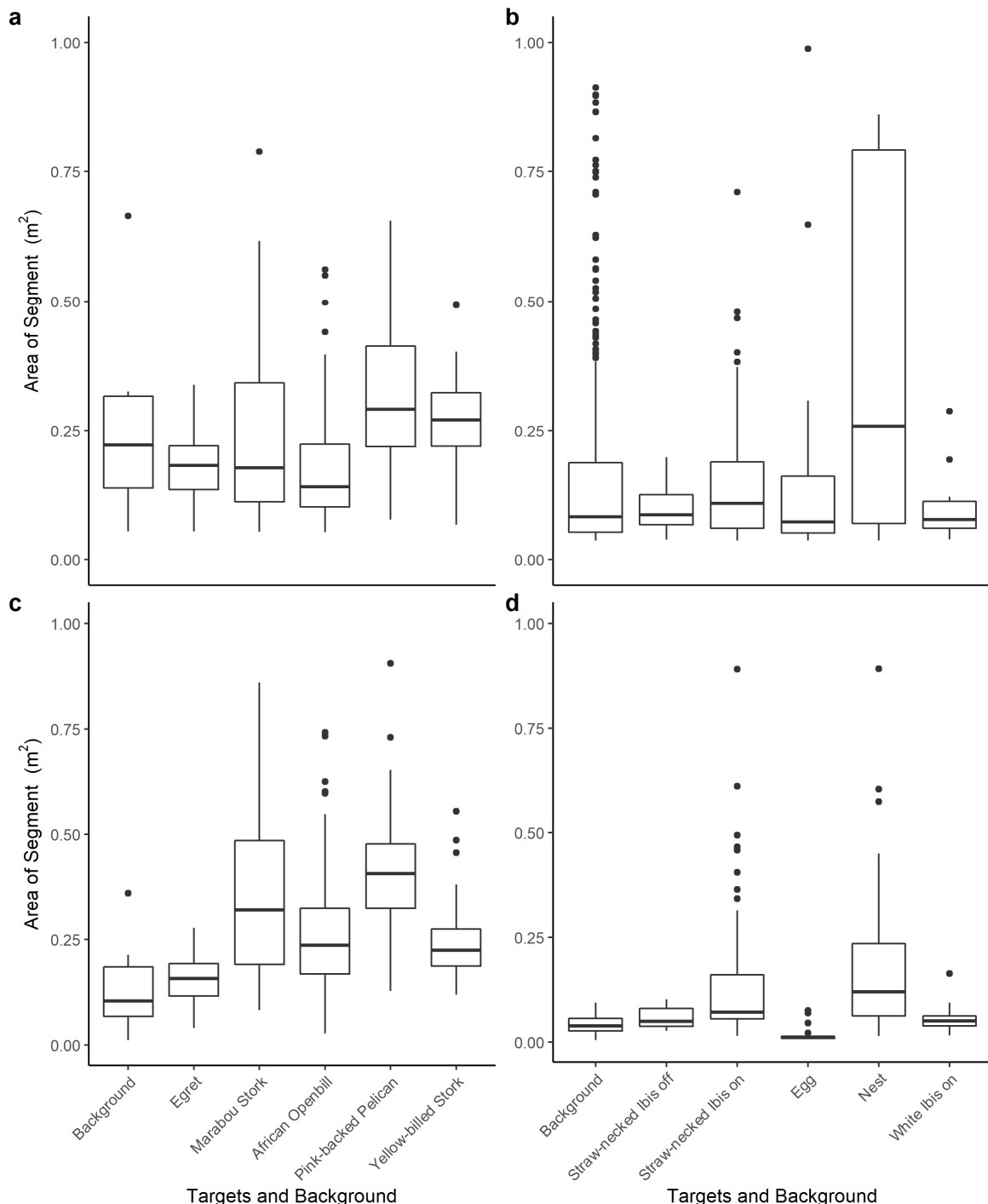

**Figure 4.** The boxplot used to identify classification errors between segment areas of targets and background using 99% thresholding for the freeware method at (**a**) the Kanana colony and (**b**) the Eulimbah colony, and the payware method at (**c**) Kanana and (**d**) Eulimbah. At the Eulimbah colony, birds were identified as being either on or off their nests.

### 2.3.4. Estimation of Target Populations

Once classifications were cleaned, we could estimate numbers (i.e., targets) for each species in the Kanana colony, summing the semi-automated classifications, given limited clumping in this colony (Figure 2, Step 10). In contrast, birds in the Eulimbah colony often nested closely together, demanding an additional step for estimation of numbers, as our classification inevitably segmented a

group of nesting birds as a single target. To estimate individual bird numbers in these clumped targets, we divided all bird classifications by average bird size (~0.08 m² [46]), before summing to estimate numbers of individuals of the two species in the colony (rounded to integer) (Figure 2, Step 10). Before estimating the nest count at Eulimbah, we filtered out other targets (e.g., empty nests) which were less than 'bird size', to remove noise and misclassifications that could not be birds or nests. To compare semi-automated count estimates across the entire image sets to the 'true' count, we also manually counted all birds in both colonies by separating the imagery into grids and summing grid numbers. We compared these estimates to our semi-automated counts, including the time taken for both counts.

## 3. Results

The Kanana colony consisted of 45 stitched images of 7,181,016 pixels (size ~ 5.5 mm), covering an area of ~39,500 m² while the stitched orthomosaic image for the Eulimbah colony had 41,785,728 pixels (size ~ 3 cm) extending over an area of ~120,000 m². It took 650 and 250 min for respective total manual counts of the Kanana and Eulimbah colonies. In comparison, our semi-automated approach took 480 min for initial development and an additional 60 min to edit the code for the Eulimbah colony. This was a time savings of about 26%, including the development of this method. Excluding this development, we estimated that about 90 min was required to work through the ten steps (Figure 2), an estimated time savings of 250–700% (not including processing time, given this can occur independently on the computer, and would differ between systems). In the Kanana imagery, we manually counted 4140 birds from five species, while Eulimbah had 3443 birds from two species, including nests totalling 6310 targets (Table 2).

**Table 2.** Final target counts for both the Kanana and Eulimbah colonies with calculations of manual versus semi-automated methods.

| Colony | Target | Final Counts | | | Difference % | |
|---|---|---|---|---|---|---|
| | | **Freeware** | **Payware** | **Manual** | **Freeware** | **Payware** |
| Kanana | Bird [1] | 2128 | 1797 | | | |
| | Egret Sp. [2] | 587 | 605 | 578 | 1.56 | 4.67 |
| | Marabou Stork | 156 | 102 | 137 | 13.87 | −25.55 |
| | African Openbill | 725 | 681 | 2986 | −4.45 [3] | −17.01 [4] |
| | Pink-backed Pelican | 154 | 71 | 59 | 161.02 | 20.34 |
| | Yellow-billed Stork | 336 | 354 | 380 | −11.58 | −6.84 |
| | Total targets | 4086 | 3610 | 4140 | −1.30 | −12.80 |
| Eulimbah | Bird [1] | N/A | 1155 | | | |
| | Egg | 108 | 287 | 80 | 35.00 | 258.75 |
| | Nest | 3458 | 3390 | 2787 | 24.08 | 21.64 |
| | Straw-necked Ibis on nest | 2271 | 2590 | 3267 | −30.49 | −20.72 |
| | Straw-necked Ibis off nest | 196 | 91 | 136 | 44.12 | −33.09 |
| | White Ibis on nest | 111 | 99 | 40 | 177.50 | 147.50 |
| | Total targets | 6144 | 7612 | 6310 | −2.63 | 20.63 |

[1] Originally, background segments which based on their probabilities were reassigned to a general 'bird' category, and upon inspection of the error matrix identified as mostly African Openbills. This step was not necessary at Eulimbah using the freeware method. [2] Predominantly Yellow-billed Egrets *Egretta intermedia* with some Great Egrets *Ardea alba*. [3] −75.72 before assigning the misclassified background segments from the 'bird' category as African Openbills. [4] −77.19 before assigning the misclassified background segments from the 'bird' category as African Openbills.

Using freeware to estimate numbers of breeding birds of each species in the Kanana and Eulimbah colonies, our initial accuracies were respectively, 88% and 99% (Table 3). In the Kanana colony, African Openbills had the lowest detection accuracy, and were likely contributing to the initial low-accuracy measure. Once we applied our probability threshold method (Figure 3) and inspected the error matrix

(Table 4), we identified that many nesting African Openbills were misclassified as background, because of their dark plumage and its similarity to the background. We corrected this misclassification by delineating background as any area with a probability (bird classification) of <0.3 or >1, (Figure 4a), producing a recalculated accuracy of 99% (Table 3). For the Eulimbah colony, it was not necessary to separate birds from backgrounds with the probability threshold method, and we only corrected for area (>0.5 as background, Figure 4b), producing a final bird detection accuracy of 98% (Table 5). Finally, after these corrections, our estimated counts using freeware were within 2% and 3% of respective manual counts for the Kanana and Eulimbah colonies (Table 2).

**Table 3.** Results for the freeware and payware used in the development of semi-automated counting methods for the Kanana and Eulimbah colonies, showing the initial, secondary (after correcting for probabilities) and final model accuracies (after correcting for area).

| Kanana Freeware | Initial | Secondary | Final |
|---|---|---|---|
| Target versus Background Accuracy | 0.99 | 0.99 | 0.91 |
| Between Target Detection Accuracy | 0.88 | 0.88 | 0.99 |
| **Kanana Payware** | | | |
| Target versus Background Accuracy | 0.99 | 0.99 | 0.90 |
| Between Target Detection Accuracy | 0.57 | 0.82 | 0.99 |
| **Eulimbah Freeware** | | | |
| Target versus Background Accuracy | 0.98 | N/A [1] | 0.98 |
| Between Target Detection Accuracy | 0.99 | N/A | 0.98 |
| **Eulimbah Payware** | | | |
| Target versus Background Accuracy | 0.99 | 0.99 | 0.93 |
| Between Target Detection Accuracy | 0.88 | 0.93 | 0.98 |

[1] It was not necessary to correct for bird probabilities at Eulimbah using the freeware method, hence the N/A values in the secondary model accuracies.

**Table 4.** Results for the freeware and payware used in development of semi-automated methods for the Kanana colony, showing the secondary error matrix after correcting for probabilities, and the final error matrix after correcting for area, where rows are the test data and columns are the predicted data.

| Kanana Freeware | | | | | | |
|---|---|---|---|---|---|---|
| | Background | Bird | Egret Sp. | Marabou Stork | African Openbill | Pink-Backed Pelican | Yellow-Billed Stork |
| Background | 3310 | 14 | 0 | 0 | 0 | 0 | 0 |
| Egret Sp. [a] | 0 | 0 | 11 | 0 | 0 | 0 | 0 |
| Marabou Stork | 0 | 6 | 0 | 5 | 0 | 0 | 0 |
| African Openbill | 14 | 11 | 0 | 0 | 7 | 0 | 0 |
| Pink-backed Pelican | 0 | 0 | 0 | 0 | 0 | 7 | 0 |
| Yellow-billed Stork | 0 | 1 | 2 | 0 | 0 | 1 | 10 |
| | Background | Bird | Egret Sp. | Marabou Stork | African Openbill | Pink-Backed Pelican | Yellow-Billed Stork |
| Background | 2 | 10 | 1 | 0 | 0 | 0 | 0 |
| Egret Sp. [a] | 0 | 0 | 50 | 0 | 0 | 0 | 2 |
| Marabou Stork | 0 | 4 | 0 | 49 | 0 | 0 | 0 |
| African Openbill | 3 | 12 | 0 | 0 | 126 | 0 | 0 |
| Pink-backed Pelican | 1 | 0 | 1 | 0 | 0 | 28 | 0 |
| Yellow-billed Stork | 0 | 0 | 2 | 0 | 0 | 0 | 66 |

**Table 4.** *Cont.*

| | Background | Bird | Egret Sp. | Marabou Stork | African Openbill | Pink-Backed Pelican | Yellow-Billed Stork |
|---|---|---|---|---|---|---|---|
| **Kanana Payware** | | | | | | | |
| Background | 3310 | 14 | 0 | 0 | 0 | 0 | 0 |
| Egret Sp. [a] | 0 | 0 | 11 | 0 | 0 | 0 | 0 |
| Marabou Stork | 0 | 6 | 0 | 5 | 0 | 0 | 0 |
| African Openbill | 14 | 11 | 0 | 0 | 7 | 0 | 0 |
| Pink-backed Pelican | 0 | 0 | 0 | 0 | 0 | 7 | 0 |
| Yellow-billed Stork | 0 | 1 | 2 | 0 | 0 | 1 | 10 |
| | **Background** | **Bird** | **Egret Sp.** | **Marabou Stork** | **African Openbill** | **Pink-Backed Pelican** | **Yellow-Billed Stork** |
| Background | 2 | 10 | 1 | 0 | 0 | 0 | 0 |
| Egret Sp. [a] | 0 | 0 | 50 | 0 | 0 | 0 | 2 |
| Marabou Stork | 0 | 4 | 0 | 49 | 0 | 0 | 0 |
| African Openbill | 3 | 12 | 0 | 0 | 126 | 0 | 0 |
| Pink-backed Pelican | 1 | 0 | 1 | 0 | 0 | 28 | 0 |
| Yellow-billed Stork | 0 | 0 | 2 | 0 | 0 | 0 | 66 |

[a] Predominantly Yellow-billed Egrets *Egretta intermedia* with some Great Egrets *Ardea alba*.

**Table 5.** Results for the freeware and payware used in development of semi-automated methods for the Eulimbah colony, showing the secondary error matrix after correcting for probabilities (not necessary at Eulimbah using the freeware method), and the final error matrix after correcting for area, where rows are the test data and columns are the predicted data.

| | Background | Bird [1] | Egg | Nest | Straw-Necked Ibis On Nest | Straw-Necked Ibis Off Nest | White Ibis On Nest |
|---|---|---|---|---|---|---|---|
| **Eulimbah Freeware** | | | | | | | |
| Background | 366 | N/A | 0 | 1 | 0 | 0 | 0 |
| Egg | 2 | N/A | 19 | 3 | 0 | 0 | 0 |
| Nest | 2 | N/A | 0 | 194 | 1 | 0 | 0 |
| Straw-necked Ibis on nest | 4 | N/A | 0 | 3 | 162 | 0 | 0 |
| Straw-necked Ibis off nest | 0 | N/A | 0 | 0 | 1 | 21 | 0 |
| White Ibis on nest | 0 | N/A | 0 | 1 | 0 | 0 | 19 |
| **Eulimbah Payware** | | | | | | | |
| | **Background** | **Bird** | **Egg** | **Nest** | **Straw-Necked Ibis On nest** | **Straw-Necked Ibis Off nest** | **White Ibis On nest** |
| Background | 1243 | 0 | 0 | 2 | 2 | 0 | 0 |
| Egg | 0 | 1 | 3 | 1 | 0 | 0 | 0 |
| Nest | 4 | 1 | 0 | 31 | 0 | 0 | 1 |
| Straw-necked Ibis on nest | 1 | 1 | 0 | 2 | 28 | 1 | 0 |
| Straw-necked Ibis off nest | 1 | 1 | 0 | 0 | 0 | 2 | 0 |
| White Ibis on nest | 0 | 0 | 0 | 0 | 0 | 0 | 4 |
| | **Background** | **Bird** | **Egg** | **Nest** | **Straw-Necked Ibis On nest** | **Straw-Necked Ibis Off nest** | **White Ibis On nest** |
| Background | 1 | 2 | 0 | 2 | 0 | 0 | 0 |
| Egg | 1 | 0 | 22 | 1 | 0 | 0 | 0 |
| Nest | 0 | 0 | 0 | 31 | 0 | 0 | 0 |
| Straw-necked Ibis on nest | 3 | 2 | 0 | 0 | 111 | 0 | 0 |
| Straw-necked Ibis off nest | 0 | 3 | 0 | 0 | 0 | 19 | 0 |
| White Ibis on nest | 0 | 0 | 0 | 1 | 0 | 0 | 19 |

[1] It was not necessary to correct for bird probabilities at Eulimbah using the freeware method, hence the N/A values.

Using payware, our initial bird detection accuracies for the Kanana and Eulimbah colonies respectively, were 57% and 88%. After re-classifying bird and backgrounds, based on the probability boxplot (<0.1 as background and >0.2 as birds for Kanana and <0.5 as background and >0.5 as birds for Eulimbah), we improved the accuracy to 85% and 93%. We then re-classified using our area threshold (>1 and <0.3 as background for Kanana and <0.01 and >0.8 as background for Eulimbah). This improved respective accuracies to 99% and 99% (Tables 3 and 4). Finally, after these corrections, our estimated counts using payware were within 13% and 21% of respective manual counts for the

Kanana and Eulimbah colonies (Table 2). Using the freeware method provided a more accurate overall count compared to the total manual counts than using payware (Table 2).

The different steps (Figure 2) had an associated code within R for freeware and payware, allowing modification and transfer from the Kanana colony where it was developed to the Eulimbah colony. Alteration in the code between colonies is firstly in the correct usage of target object names (which naturally differ based on the species or object being counted, Figure 2, Step 5). Secondly, thresholds used to differentiate between and re-classify targets will differ based on the segment statistic used and the target objects' physical attributes (e.g., area or colour, Figure 2, Step 9). The major alteration to code required when transferring between freeware and payware is assigning the correct predictor variables to the random forest modelling, based on the output of the image statistics of each segment (Figure 2, Step 3). All code/data required are available for download (see Table S1).

## 4. Discussion

Methods which can rapidly collect data over large areas and process these data quickly are important for understanding systems and in providing timely data analyses to managers and the public. Drones are becoming increasingly powerful tools for the collection of such data on a range of organisms [25,47,48], given that they can capture imagery over inaccessible and sometimes dangerous areas. This is only half the process: the imagery needs to be analysed to provide estimates of organisms. Manual counting is the traditional approach but, it is slow and laborious and may be prone to error. New and improved methods are required to process images quickly and efficiently. Our semi-automated system of counting breeding waterbirds on nests on highly variable backgrounds was effective and efficient. Importantly, we successfully applied the methodology, developed on one colony with different species in a different environment (on another continent) to another colony. This transferability is particularly useful. Significantly, payware and freeware methods were equally effective and accurate, providing global opportunities where resourcing is limited. Finally, there are opportunities to apply this approach to other organisms, amassing in large aggregations.

Using our approach, waterbird colonies around the world could be quickly and accurately counted using drone data. There are many active research teams, often providing information for management, surveying and estimating sizes of breeding colonies of waterbirds, including colonies in Australia [49], Southern India [50] and Poland [51]. But our methodology is also transferable to other aggregations of species, such as the Valdez elephant seal *Mirounga leonine* colony in Patagonia [52] or macaques *Macaca fuscata* in tourist areas in Japan [53]. Transferability requires some key idiosyncratic steps in image processing, data training and modelling. These include either the initial clipping of overlap in drone imagery or the creation of orthomosaics, then the development of a training model for classifying species (Figure 2, Step 2) and finally, testing the model using derived thresholds (Figure 2, Step 9), discriminating between animals and backgrounds. Such steps can be applied to drone imagery captured in different environments, making the use of citizen science drone-collected imagery a feasible data source [54].

Every species of waterbird or other organism will differ in some way from the background, be it in size, colour or a combination of multiple such image statistics. To edit and implement our methodology for any waterbird colony around the world, after initial image processing, the manually annotated dataset must be created to train the model on target species. Subsequently, edits must be made to the R code aligning target names and the image statistics to be used as predictors, which can then be used to estimate thresholds distinguishing species from backgrounds. Extending to other organisms can take a similar approach, with final modelling dependant on the creation of the initial manually annotated dataset classifying the organisms and background. While each study will have its own requirements for the data, we aimed to develop a methodology that would produce a maximum of 10% disparity between semi-automated and manual counts, which with more time invested could be further reduced.

Consideration of drone height is an important first step when collecting imagery for use with this method. In general, a lower flight height and a better camera will produce images of a larger pixel

size, however this needs to be balanced against disturbance to the species of interest. Furthermore, a lower drone height equates to less area covered in the imagery within the span of one battery, and so the number of available batteries and survey area therefore need to be considered. When surveying a single species that contrasts a relatively simple background, less image detail will be required to differentiate between the target and background. Conversely, the more species to differentiate between, particularly if on a varied background such as the colony at Kanana, the more detail required in the imagery to obtain accurate estimates. Drone height requirements will therefore be unique to study location, area, species and aims.

The most challenging aspect of our methodology was identifying and dealing with misclassification errors. Ultimately, inaccuracy occurs and needs to be reported. Identifying the source of errors is critical and there are two ways to improve the final estimates: increasing sample sizes of training data and identifying attributes that better discriminate between objects and backgrounds. Increasing sample sizes of training datasets likely improves models. This may be particularly relevant where colonies are repeatedly surveyed (i.e., multiple breeding events over time), as the greater initial time investment in training the model may reduce the time required for following surveys. We only used ~50 individual objects for each species' grouping, which may have reduced the power of our models. For example, for the pink-backed pelicans in the Kanana imagery, we only had 32 training points (as they were relatively rare in the colony) and so increasing sample size in future years or from other sources would probably improve the model and classification. Increased sample sizes are particularly important for discriminating between highly similar target objects, improving the model's discriminatory capacity to identify a unique attribute or set of attributes for each object.

Even with reasonable sample sizes, there may be confusion among species and the background, contributing to errors. For our Kanana colony, the dark plumage of the African Openbills was often confused with dark patches of background, such as water. Also, similarly sized, shaped and coloured egret species could be confused with Yellow-billed Storks, contributing to inaccuracies (Table 2). As well as size, there could be other sources of discrimination between targets (e.g., pigmentation means or deviations) which could be incorporated in modelling and identified from boxplots (Figure 2, Step 9). Our script can easily be altered, to incorporate such a change. Improvements in image collection such as the use of a multi-spectral sensor (as opposed to the combined standard Red Green Blue sensor used here) could also improve modelling and separation of backgrounds from birds. Further, software improvements could also improve outcomes. Inevitably, more data, repeated measurements and time invested will improve effectiveness, accuracy and efficiency, in the equally performing freeware and payware software (Table 2).

There were considerable time efficiency benefits in using our semi-automated approach. We differentiated among five species in 26% less time than when we used manual counting, with time savings likely to improve with repeated counts due to user experience. Further, such manual counting was probably also prone to some error, as observers tire or inadequately discriminate. Increasingly, machine learning approaches are improving and becoming more accurate than manual methods in a range of disciplines (e.g., medicine, identification of melanomas [55] and astronomy, identification of chorus elements [56]). There is no reason why our approach, and more broadly, approaches of counting animals using drone imagery and machine learning, will not become increasingly more accurate and more efficient with growing amounts of data, with wide applications. Such savings in time would allow for counts and reports to be rapidly provided to environmental managers, providing information for real-time management decisions, where field data may not be sufficient [29].

Drone imagery can also provide baseline data of environmental importance. Although the Kanana colony is one of the biggest and most frequently used breeding grounds of waterbirds in the Okavango Delta, a United Nations Educational Scientific and Cultural Organization (UNESCO) World Heritage Site, there are few quantitative data on the size or composition of this breeding colony. Another six colonies in the Okavango Delta similarly have little information. Some of these are difficult and dangerous (crocodiles, hippopotamuses, elephants) to approach on foot and so drones provide an

excellent data collection method. The importance of these data could grow when combined with increasing knowledge of the link between flooding and waterbird breeding. Similarly, the Eulimbah colony is one of the only breeding colonies of up to 50,000 breeding pairs in the Lowbidgee floodplain, which also includes other breeding species, such as cormorants, herons and spoonbills. These data are also increasing in their value in determining historical impacts of river regulation and management on major wetlands [57,58], as well as guiding management of flows to improve waterbird breeding and wetland condition [59,60].

The use of drones and the processing of imagery for ecological applications will increase, given their advantages. Processing methods also continue to improve to capitalise on this technology, increasing our understanding and ability to manage complex ecosystems, not only for waterbird colonies but other aggregations of wildlife. Eventually, software informed by training data could be installed on drones, allowing real-time processing and estimation of numbers of birds or other target organisms. Until this happens, the semi-automated methods described here provide considerable promise and opportunity around the world, with the added values of efficiency, free software options and opportunity for improvements in accuracy.

## 5. Conclusions

We developed a semi-automated machine learning counting method, using both freeware and payware, that was transferable between waterbird colonies on different continents. Our detection accuracy (targets), between the training and test data, was 91% for the Okavango Delta colony and 98% for the Lowbidgee floodplain colony. These estimates were within 1–5%, whether using freeware or payware for the different colonies. Our semi-automated method was 26% quicker, including development, and 500% quicker without development than manual counting. Using drones and semi-automated counting techniques therefore saves time and resources, whilst allowing access to difficult or dangerous areas. As a result, the use of drones as scientific tools will increase, particularly to survey wildlife aggregations. Importantly, their low cost and the option of using freeware provides research opportunities globally, including where resourcing is limited. We predict that these benefits will only increase as battery life is extended and a greater range of drone software options become available.

**Supplementary Materials:** The following are available online at http://www.mdpi.com/2072-4292/12/7/1185/s1, Figure S1 and Table S1 can be found in the Supplementary Materials section.

**Author Contributions:** Conceptualization, R.J.F., M.B.L. and R.T.K.; Data curation, R.J.F. and M.B.L.; Formal analysis, R.J.F. and M.B.L.; Funding acquisition, R.J.F., K.J.B. and R.T.K.; Investigation, R.J.F. and M.B.L.; Methodology, R.J.F. and M.B.L.; Project administration, K.J.B.; Software, R.J.F. and M.B.L.; Supervision, K.J.B. and R.T.K.; Writing—original draft, R.J.F.; Writing—review and editing, R.J.F., M.B.L., K.J.B. and R.T.K. All authors have read and agreed to the published version of the manuscript.

**Funding:** This research received financial support from Elephants without Borders, Taronga Conservation Society, the Australian Commonwealth Environmental Water Office, the NSW Department of Primary Industries, and the University of New South Wales Sydney.

**Acknowledgments:** We thank the New South Wales Department of Primary Industries for providing access to the Eulimbah property, and similarly we thank Ker & Downey Kanana Camp for access to the Kanana colony. We acknowledge Max Phillips for his role in flying the drone at the Eulimbah colony. This study was conducted under the guidelines of the UNSW Animal Care and Ethics, permit 13/3B. We also thank the Government of Botswana for access to research permits EWT 8/36/4 XXIV (179), and drone permit RPA (H) 211.

**Conflicts of Interest:** The authors declare no conflict of interest. The funders had no role in the design of the study; in the collection, analyses, or interpretation of data; in the writing of the manuscript, or in the decision to publish the results.

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
