# Peer review of "Counting Mixed Breeding Aggregations of Animal Species Using Drones: Lessons from Waterbirds on Semi-Automation"

_remotesensing, doi:10.3390/rs12071185_

Round 1
Reviewer 1 Report
This manuscript has been prepared well, with the script being made freely available. My only concern is that it draws in a wide enough audience, and I have suggested how to achieve this in the first paragraph. I have only a couple of minor comments. I look forward to seeing this work published.
Line 14, there is something wrong with the English in this sentence, please check, maybe “are also allowing?”
Line 31, I would advise not limiting it to sea birds immediately, draw on the wide range of species being monitored with drones in aggregations, giving examples from different groups.
Line 33 – why only breeding colonies, why not also wintering/foraging colonies?
Reviewer 2 Report
Thank you for the opportunity to review this paper which I found to be well written and very interesting. The mapping of bird numbers using drones can assist in bird counts and potentially be rolled out to other species so the potential of this method for application in conservation science is good.
I have a few minor comments which should be addressed before the paper is accepted for publication.
Ln 75. It looks like you have 4 objectives and one point of discussion.
Figure 1: It may be tricky but it will really help the reader who is used to more traditional mapping and remote sensing methods to have a scale bar perhaps on bottom two images. It is also strange to see the segmentation presented on a figure before the method is explained. Perhaps you can remove the segmentation and just use arrows to point to the birds?
Ln 122: Why did you use different image capturing techniques for your two different sites? And what are the potential implications (if any) of this? The height the drone flies has an impact on pixel resolution. This will in turn impact on the heterogeneity of the image. Do you have any recommendations for others wanting to conduct this research on optimal height to fly? How do you make these decisions?
Figure 3 & 4: I presume the on and off refers to on and off a nest? Just make a note of this in the caption.
Table 2: I have a concern with the category “Bird”. I think it might be better to present the results in two level of detail. One showing how accurately the method detects bird or no bird. Second report on how accurately each method detects the correct species. I think this will make the research more useful for those making a decision on which method to choose should they want to do similar research.
Table 3: This table is difficult to follow because you have too much information on it. Consider breaking in to two separate tables.
Table 4: See comment for Table 3.
Discussion: I am not sure how much value the time assessment adds to the paper. For me it could depend on factors such as experience of the operator. I think the paper stands well without this part but it is not make or break so I am happy to leave it up to the authors in light of comments from other reviewers. I do think that the discussion should include a paragraph on the height the drone flies, how this relates to pixel resolution and in turn the heterogeneity of the scene captured. I think this is important information which is missing and as someone who has done a lot of remote sensing work but never with drones, it would be really helpful to have this included. I am sure I am not the only reader who will find it helpful.
Conclusion: Can you expand on this? What are the main findings rather than the rather dry figures which you report? Do you have any recommendations for further research?
